# Applications of Molecular Imprinting Technology in the Study of Traditional Chinese Medicine

**DOI:** 10.3390/molecules28010301

**Published:** 2022-12-30

**Authors:** Yue Zhang, Guangli Zhao, Kaiying Han, Dani Sun, Na Zhou, Zhihua Song, Huitao Liu, Jinhua Li, Guisheng Li

**Affiliations:** 1School of Pharmacy, Collaborative Innovation Center of Advanced Drug Delivery System and Biotech Drugs in Universities of Shandong, Key Laboratory of Molecular Pharmacology and Drug Evaluation (Yantai University), Ministry of Education, Yantai University, Yantai 264005, China; 2CAS Key Laboratory of Coastal Environmental Processes and Ecological Remediation, Shandong Key Laboratory of Coastal Environmental Processes, Shandong Research Center for Coastal Environmental Engineering and Technology, Yantai Institute of Coastal Zone Research, Chinese Academy of Sciences, Yantai 264003, China; 3College of Chemistry and Chemical Engineering, Yantai University, Yantai 264005, China

**Keywords:** molecular imprinting technology (MIT), molecularly imprinted polymers (MIPs), Traditional Chinese Medicine (TCM), active components, hazardous components

## Abstract

Traditional Chinese medicine (TCM) is one of the most internationally competitive industries. In the context of TCM modernization and internationalization, TCM-related research studies have entered a fast track of development. At the same time, research of TCM is also faced with challenges, such as matrix complexity, component diversity and low level of active components. As an interdisciplinary technology, molecular imprinting technology (MIT) has gained popularity in TCM study, owing to the produced molecularly imprinted polymers (MIPs) possessing the unique features of structure predictability, recognition specificity and application universality, as well as physical robustness, thermal stability, low cost and easy preparation. Herein, we comprehensively review the recent advances of MIT for TCM studies since 2017, focusing on two main aspects including extraction/separation and purification and detection of active components, and identification analysis of hazardous components. The fundamentals of MIT are briefly outlined and emerging preparation techniques for MIPs applied in TCM are highlighted, such as surface imprinting, nanoimprinting and multitemplate and multifunctional monomer imprinting. Then, applications of MIPs in common active components research including flavonoids, alkaloids, terpenoids, glycosides and polyphenols, etc. are respectively summarized, followed by screening and enantioseparation. Related identification detection of hazardous components from TCM itself, illegal addition, or pollution residues (e.g., heavy metals, pesticides) are discussed. Moreover, the applications of MIT in new formulation of TCM, chiral drug resolution and detection of growing environment are summarized. Finally, we propose some issues still to be solved and future research directions to be expected of MIT for TCM studies.

## 1. Introduction

Traditional Chinese Medicine (TCM) resources are a rich and valuable drug pool in China. With the development of society and economy, more and more attention has been paid to the development and utilization of TCM resources worldwide [1]. Especially after Youyou Tu from China won the Nobel Prize for her immense contributions to artemisinin, the research enthusiasm of TCM has been aroused globally [2]. TCM can be classified into various categories [3]: (1) according to drug function, such as detoxification, heat-clearing, qi-regulating and blood-stasis-activating herbs, etc.; (2) according to the classification of medicinal parts, such as roots, leaves, flowers, bark, etc.; (3) according to the classification of active components, such as herbs containing alkaloids, herbs containing volatile oils, herbs containing glycosides, etc.; and (4) according to natural properties and affinities [4]. TCM is also divided into botanical, animal and mineral medicines. Animal and plant herbs are categorized and organized according to their original plant and original animal affinities, for example, ephedraceae, magnoliaceae, buttercup family, etc.

TCM works through the various physiologically active chemicals. Most of the chemical active components of TCM are characterized by low contents, complex structures, diverse types and unstable properties [5]. The above shortcomings not only make the separation and purification of active substances more difficult, but also hinder the research of molecular level and development of related products [6]. Most TCM is complex mixtures, and the common separation and purification methods can only get crude extracts, but it is quite difficult to get a single pure component.

At present, the main separation and purification technologies of active components in TCM include extraction separation [7] (aqueous two-phase extraction, supercritical extraction [8], etc.), chromatographic separation (supercritical fluid chromatography [9], high-speed countercurrent chromatography [10], etc.), recrystallization, membrane separation [11], molecular distillation [12], macroporous resin separation [13], etc. In order to obtain high purity of active components, the chromatography separation technique and repeated recrystallization operations are most widely used. These technologies usually have disadvantages such as cumbersome operation, long separation cycle, low efficiency, large consumption of solvents and separation materials or low selectivity.

Molecular imprinting technology (MIT) is a molecular recognition technology based on the principle of the interaction between antigens and antibodies [14]. The prepared molecularly imprinted polymers (MIPs) have the three major features of structure predictability, recognition specificity and application universality [15,16]. Since the pioneering work of Polyakov [17], MIT and MIPs have shown good application and development prospects in chiral drug resolution [18], chemical/biological sensors [19,20], simulated antigens and antibodies [21], simulated enzyme catalysis, natural product separation [22,23,24], environmental analysis [25] and other fields. In addition, MIPs are easy to prepare and can be reused, making them one of the most widely concerned technologies for the separation and extraction of the effective components of modern TCM [26,27]. In Table 1, the advantages and disadvantages of several separation methods are compared [7,9,10,11,12,13,14,23,24,25].

According to the literature review, there are few related reports on MIT for TCM study before 2017. Most reports have been published since 2017. Therefore, in this work, the applications of MIT in the study of TCM are reviewed comprehensively, focusing on the recent advances since 2017. Firstly, traditional polymerization methods for MIPs; emerging technologies represented by surface imprinting and nanoimprinting; and new strategies represented by multitemplate, multifunctional monomer and dummy template imprinting are introduced. Then, the research progress of MIPs in purification enrichment and detection of active components, analysis of hazardous components and other applications for TCM are highlighted. Finally, the further perspectives of MIPs in TCM are prospected.

## 2. Methods, New Imprinting Technologies and Strategies for Preparing TCM-Related MIPs

At present, besides traditional methods for preparing TCM-related MIPs, a variety of new methods have been developed. The polymerization methods and new imprinting technologies and strategies are introduced as follows.

### 2.1. Polymerization Methods for TCM-Related MIPs Preparation

The polymerization methods include bulk polymerization, precipitation polymerization, suspension polymerization and others, which are selected based on the desired characteristics of the required MIPs and the template [28].

#### 2.1.1. Bulk Polymerization

Bulk polymerization is one of the most frequently used and the simplest polymerization methods for MIPs [29]. The functional monomers and the templates are mixed before adding the crosslinker and initiator. Oxygen is purged from the system by bubbling with an inert gas like nitrogen, and then the polymerization process is initiated by thermal or photo initiation. The method involves forming a large-sized MIPs followed by grinding into smaller particles with irregular shapes and heterogeneously distributed binding sites. Some of these particles may also have minor or no binding sites [30]. Bulk polymerization has been used to prepare MIPs for the extraction of salidroside from rhodiola rosea [31], lappaconine from aconitum [32] and formononicin from red clover [33].

#### 2.1.2. Precipitation Polymerization

Precipitation polymerization is a one-step polymerization method that is known for high yield of uniform spherical particles [34]. The method depends on the saturation phenomena that a great amount of the template is mixed with a diluted monomer solution. When polymerization occurs, the MIPs slowly precipitate from the solution. High amounts of solvent and template consumption are the main drawbacks of this method. Moreover, the reaction temperature and solvent polarity need to be controlled carefully to obtain the required particle size. Precipitation polymerization has been increasing used for preparing MIPs for the extraction of ginsenoside Rg_1_ from ginseng [35], matrine from Sophora alopecuroides [36] and resveratrol from poria cocos [37].

#### 2.1.3. Suspension Polymerization

Suspension polymerization is also a one-step technique that the insoluble monomer is dispersed dropwise under vigorous stirring with a stabilizer to produce relatively big-sized spherical particles [38]. Mineral oils, perfluorocarbon and water can be used as the continuous phase. Volume rate and stirring speed have a direct influence on the particles’ sizes. Suspension polymerization has been used for the synthesis of MIPs to extract aristolocholic acid I (AAI) from Caulis Clematidis Armandii [39] and transform the drug dosage form of oral huperzine A [40].

#### 2.1.4. Sol–gel Polymerization

The sol–gel polymerization method is a relatively simple method to synthesize MIPs. The compounds containing highly chemically active components as precursors are mixed uniformly in the liquid phase, then form a stable transparent sol system through hydrolysis and condensation chemical reactions to form a stable transparent sol system in solution. The sol is slowly polymerized between the aged particles to form a three-dimensional network structure. Then the gel network is formed by filling with solvents losing fluidity, which are dried, sintered and cured to prepare a material with molecular and even nanostructures [41]. This sol–gel polymerization can perform in aqueous phase under the mild conditions (such as room temperature), which overcomes the shortcomings of the abovementioned free radical polymerization and has aroused wide interest. For example, the polymerization has been used to attain the MIPs for the extraction of celastrol from tripterygium [42].

### 2.2. New Imprinting Technologies and Strategies for TCM-Related MIPs Preparation

The traditional preparation methods of MIPs are relatively simple, but the synthesized MIPs have relatively fewer recognition sites. Nowadays, the core-shell structured MIPs have been developed to improve the adsorption capacity and increase the effective recognition sites. The synthesis process of MIPs can effectively improve the performance of MIPs by using surface imprinting and nanoimprinting technology and multitemplate, multifunctional monomer, dummy template and stimulus response imprinting strategies. Figure 1 shows the several new imprinting technologies and strategies for preparing TCM-related MIPs.

#### 2.2.1. New Imprinting Methods

##### Surface Imprinting Technology

Surface imprinting technology refers to preparing imprinted layer on the surface of the carrier through modification [44]. MIPs are prepared more easily by grafting some active groups onto silica gel microspheres, carbon nanotubes, alumina or titanium oxide, organic polymers, chitosan and other substrates. Surface molecular imprinting mainly adopts self-assembly, sol–gel polymerization, sacrificial silica gel skeleton method, graft copolymerization, hydrolytic polycondensation and chemical vapor deposition methods. The prepared polymers do not need grinding and other post-processing, and the microspherical MIPs can be directly obtained, which better preserves the integrity of the polymers [45]. Surface-imprinted polymers greatly shorten the elution time of template molecules and improve the adsorption capacity, which can achieve adsorption equilibrium in a shorter time. However, due to the existence of binding sites only on the surface of the polymer material, the polymer’s binding capacity obtained by surface imprinting technology is low [46]. Surface imprinting technology has been widely used for the synthesis of ginsenoside Re [47], AAI [48] and cyfluthrin [49] templated MIPs.

##### Nanoimprinting Technology

Nanomprinting technology means preparing MIPs with nano-structural dimensions [50]. The nano-MIPs do not need additional procedures like grinding, screening, etc. and possess high mechanical strength, which means that the recognition sites are not easy to damage. The nanoscale MIPs show a large specific surface area, and template molecules can be almost completely removed, which can maximize the proportion of effective binding sites. Most of these binding sites are located on or near the surface of materials, showing high binding capacity. Meanwhile, template molecules are easier to access the molecularly imprinted sites, thus showing fast binding dynamic characteristics. Therefore, it has wide application prospects. Moreover, the synthesis of more stable nanomaterials can play a certain role in maintaining the stability of molecularly imprinted sensors [51]. Nanoimprinting technology has been increasingly used for the synthesis of ginsenoside Re [47], celastrol [42] and calycosin [31] templated MIPs.

#### 2.2.2. New Template Strategies

##### Multitemplate Imprinting Strategy

The multitemplate imprinting strategy uses multiple targets or species as templates to generate multiple types of binding sites in one polymeric material [52]. The prepared multitemplate MIPs (mt-MIPs) can realize simultaneous extraction, enrichment and separation, and thereby analytical detection of multiple targets, which improves the MIPs’ utilization rate and shortens experimental time. It should also be noted that the selectivity of mt-MIPs (for a template molecule) is lower than that of single-template MIPs, probably due to the dilution of the number of binding sites for each template and the increased remixing effect of multiple templates [13]. In fact, the imprinting effect of mt-MIPs is the result of the balance and compromise of the imprinting properties of various template molecules [13,53]. As an ideal choice for multiresidue analysis, mt-MIPs have been also employed in the extraction and analysis of TCM components [48].

##### Dummy Template Imprinting Strategy

At present, the most widely used method to solve the template leakage problem is the dummy template imprinting strategy; that is, a molecule similar to the target molecule in structure, shape and size is used as the template for imprinting, which not only maintains the specific recognition of MIPs, but also avoids the interference caused by template molecule leakage [54]. Importantly, the same strategy applies to template molecules that are expensive, chemically unstable, prone to degradation during polymerization or have safety problems. However, it is not easy to find the right virtual template molecules in terms of chemical structure, molecular size and so on [46]. The dummy template MIPs (DMIPs) have been used for the extraction of pesticides in cinchona [55].

#### 2.2.3. New Monomer Strategies

##### Multifunctional Monomer Strategy

Multifunctional monomer imprinting strategy is to improve the non-covalent binding between the target molecule and functional monomer by using more than two functional monomers to interact with template molecule [44]. The use of multifunctional monomers is a good selection to improve the selectivity, and thereby the adsorption capacity, of MIPs. Indeed, the use of multifunctional monomers is a good way to improve selectivity and also an effective way to imprint various analytes. However, how to properly select and reasonably combine the existing multiple functional monomers, how to skillfully design and synthesize new functional monomers and how to further utilize their synergistic effects need to be explored continuously [51]. The multifunctional monomer strategy has been used for the synthesis of polyphenolic compounds MIPs [33].

##### Stimulus Response Imprinting Strategy

Stimulus response imprinting is a technique that utilizes the nature of MIPs (SR-MIPs) itself to make a keen response under special external stimulus conditions (magnetic stimulation, thermal stimulation, optical stimulation, pH, etc.) [44]. Through regulating the stimulus, the SR-MIPs can smartly control the binding and release of templates. It can achieve specific controlled release of target molecules under external stimulus conditions. At the same time, there are several major challenges and opportunities for SR-MIPs: designing and synthesizing new response functional monomers and exploring new stimulus response systems; in the development of multi-response SR-MIPs, it is necessary to pay attention to the clever and reasonable combination to ensure the best synergistic effect of multi-response elements [50]. SR-MIP-based sensors have been constructed for sensing analysis of formononicin [33] and acetylcholine [56].

## 3. Applications of MIPs in TCM Study

Nowadays, TCM has been increasingly used to treat related diseases in clinical practice, and therefore, it is imperative to further study TCM. MIPs can be used to identify and enrichment of related components in TCM, combing with analytical technologies like HPLC and sensors, which could provide simple and efficient methods to determine the contents of active and hazardous components in TCM [57]. The two major applications are the purification enrichment and determination of active components and analysis of hazardous components. Thus, the quality of TCM can be controlled. Herein, MIPs act as the selective adsorbents for solid-phase extraction (SPE), the selective stationary phase for chromatographic separation and the selective recognition and transduction elements for sensing analysis [13]. The applications of MIT in TCM are summarized in Table 2 [31,32,33,35,36,37,39,40,42,48,49,50,54,56,57,58,59,60,61,62,63,64,65,66,67,68,69,70,71,72,73,74,75,76].

### 3.1. Purification Enrichment and Determination of Active Components

The applications of MIT in the separation purification and determination of active components of TCM is an active field in the research of TCM. The number of literature reports is increasing, and the research methods are also improving. The following will give related introductions according to different kinds of TCM including glycosides, alkaloids, terpenoids, steroids, flavonoids and so on.

#### 3.1.1. Glycosides

Glycosides are a group of compounds in which a sugar or a sugar derivative is linked to another non-sugar substance by a sugar end-group carbon atom, such as baicalin, hesperidin, salidroside and ginsenosides. Baicalin has significant biological activities, such as bacteriostasis, diuresis, anti-inflammatory, cholesterol lowering, antithrombotic, relieving asthma, purging fire and detoxification, hemostasis, tocolysis, antiallergy and antispasmodic properties. Bi et al. [58] synthesized polyhedral oligomeric silsesquioxanes (POSS)-hybridized molecularly imprinted monolith combining with online solid-phase microextraction high-performance liquid chromatography (SPME-HPLC) and offline LC-MS/MS to determine baicalin and its analogues from scutellaria baicalensis by microwave-assisted extraction. The preparation steps are schematically shown in Figure 2. Baicalin can be effectively enriched in complex substrates with limit of detections (LODs) as low as 1 ng/mL. This study provides a new perspective and method for controlling the broad selectivity of MIPs and a simple two-dimensional HPLC system for the analysis of TCM.

Hesperidin is a key indicator of preparing some herbal medicines and Chinese medicinal, and it is important to achieve its sensitive, rapid and highly selective detection. Sun et al. [59] used ultrafine activated carbon as the base modification material of the electrode, and then prepared MIPs on the electrode by using self-assembly method and electropolymerization method with hesperidin as the template molecule and o-aminothiophenol as the functional monomer. Under the optimized conditions, the prepared MIP-based electrochemical sensors exhibited a wide linear range from 85 nM to 30 μM, with LODs as low as 45 nM. This is the first time that ultrafine activated carbon has been used as a carrier for MIPs. The MIP/nanoporous carbon (MIP/NC) sensor could accurately determine calycosin in the actual sample. Moreover, this study will provide a prospective exploration for the application of electrochemical analysis technology in the active component detection of TCM.

Salidroside (SD) is one of the bioactive compounds found in rhodiola rosea. Yu et al. [60] prepared a highly selective MIPs via bulk polymerization for the extraction and preconcentration of SD. The application of the developed MIPs as selective sorbents for SPE of SD was also investigated. A rapid, cost-effective and efficient method based on MIPs-SPE-HPLC was developed for the determination of SD in different parts of rhodiola rosea under optimal conditions. The method showed satisfactory recoveries of 88.74–97.64% with relative standard deviations (RSDs) ranging from 2.05 to 3.54%. The concentrations of SD in the petals of rhodiola crenulata were firstly validated by this method. MIP-SPE can be used as a useful tool for detecting and quantifying SD in a variety of TCM.

Ginsenoside is a large family of triterpenoid saponins from panax ginseng with various important biological functions. It is crucial to develop effective sample treatment and detection approaches for qualitative and quantitative analysis of ginsenosides [47]. 

Zhao et al. [48] analyzed ginsenoside content in ginseng samples by plasmonic immuno-sandwich assay with dual boronate affinity nanoparticles. The preparation method and principle of plasma immuno-sandwich assay of bis-borate affinity nanoparticles are shown in Figure 3. The Raman signal intensity of the whole system was enhanced, and the sensitivity of the method was improved by the slide coated with surface-imprinted Au nanoparticles. For real-sample applications, successful quantitative analysis of ginsenoside Re in ginseng was performed. This dual boronate affinity nanoparticle-based plasmonic immuno-sandwich assay holds great promise for many purposes, such as pharmaceutical analysis.

Liu et al. [35] developed and compared two molecular imprinting synthesis methods for ginsenoside Rg_1_. They used the same Rg_1_ as templates to synthesize MIPs by precipitation polymerization and surface imprinting, respectively. By static and kinetic adsorption experiments on two kinds of MIPs, it was proved that this method can be used for rich set separation of ginsenoside Rg_1_. This provides an experimental basis and method reference for the study of ginsenoside MIPs.

#### 3.1.2. Alkaloids

Alkaloids are a group of nitrogenous alkaline organic compounds found in nature with a complex ring structure that mostly contains the nitrogen element in it. Alkaloids have alkali-like properties and have significant biological activities, making them one of the most important active components in TCM [77].

Caffeine is an alkaloid derived from methylxanthine, which exists in tea, cocoa bean, coffee bean and other plants. A moderate dose of caffeine has certain analgesic and headache relieving effects. However, excessive intake of caffeine will cause a series of adverse effects on human health. Due to the high water solubility of caffeine, the biodegradation of caffeine into water after human metabolism lasts a long time, so caffeine also exists in environmental water samples. The reported caffeine detection methods mainly include chromatography and electroanalysis. These techniques have high sensitivity and accuracy, but selective enrichment procedure is still necessary.

Yang et al. [61] prepared magnetic MIPs (MMIPs) with a core-shell structure by using the coprecipitation method, which can selectively recognize and adsorb caffeine. The MMIPs displayed high affinity, selectivity and reusability and good thermal stabilities in the range of 0–800 °C. Adsorption studies showed that the saturated adsorption capacity of MMIPs for caffeine was 2.94 mg/g, which was much higher than that of non-molecularly imprinted polymers (NIPs). At the same time, MMIPs showed good adsorption selectivity for caffeine. In addition, the adsorption–desorption cycle experiment showed that the MMIPs possessed good reusability, which is of great significance for environmental monitoring and remediation.

Lappaconine is a diterpenoid alkaloid isolated from the root of Aconitum. Lappaconine is commonly used as an analgesic and has a strong analgesic effect, which is 7 times that of the common analgesic aminopyrin [32]. Its analgesic effect is similar to that of demerol, but the analgesic effect lasts longer. It has no addiction, no teratogenic effect and no accumulation poisoning. The liver and kidney function and hematopoietic function were not affected. At present, the separation and extraction of lappaconine from the root of Aconitine is still a relatively tedious and time-consuming work. Finding an efficient and low-cost separation method has become the focus of research.

Zhang et al. [32] quickly prepared rabaconine MIPs microspheres by bulk polymerization. The selective dissociator α of lappaconine and β-glusterol was 2.18. The synthesized polymer has good selectivity and high adsorption capacity, providing a new SPE adsorbent or chromatographic column filler for the enrichment and separation of lappaconine from natural products. This method is simple and feasible, which provides a theoretical basis for the development of a new separation and purification process of lappaconine from TCM.

Sophora alopecia is a plant of sophora pseudoacacia in leguminous family. As a kind of TCM, sophora alopecuroides mainly has the functions of clearing heat, dryness and dampness, as well as relieving pain and killing insects. The chemical constituents of sophora alopecuroides are complex, among which alkaloids are the main active component. At present, the alkaloids isolated from Sophora alopecia mainly consist of quinolidine, which can be divided into four types according to structure type: matrine-tetracyclic type, garbanine-tetracyclic type, bromine-tricyclic type and lupine-dicyclic type. Matrine alkaloids have various biological activities in neuroprotection, antitumor, antioxidant, anti-hepatitis B virus, etc.

Liu et al. [36] synthesized matrine-MIPs with good morphology and properties by precipitation polymerization and established a method for quantitative analysis of the active component groups in the extract of Sophora alopecuroides by MISPE-HPLC. The structure of the enriched components was analyzed by liquid-phase chromatography–mass spectrometry. The results showed that the imprinted polymer had special specificity and attained the adsorption equilibrium at 200 min (*Q*_max_ = 49.97 mg/g). Therefore, it can be used as a solid-phase extraction agent for the separation and enrichment of matrine and its analogues in medicinal plants. Using MIPs as SPME column filler not only greatly reduces the investment of material resources and manpower, but also further improves the efficiency and speed of screening matrine analogues. Compared with C_18_ column purification, the MISPE column can effectively remove the matrix effect, which is helpful for the separation, purification, rapid screening and determination of matrine and its analogues in medicinal plants and biological samples.

Alkaloids are strong effective components in Fritillaria, among which isosteroidal alkaloids represented by peimine, peiminine and peimisine have significant effects in the cardiovascular system and central nervous system, as well as antibacterial, antitumor and other aspects.

Han et al. [62] synthesized MIPs with high selectivity of peimine. The linear range of the method was 5.0 × 10^−7^–5.0 × 10^−5^ mol/L, and the LOD was 2.0 × 10^−7^ mol/L. The RSD of the method was 3.6% for the detection of peimine solution at 3.0 × 10^−6^ mol/L. This method has been successfully applied to the analysis and determination of fritillaria with satisfactory results, which also provides an alternative method for the separation and enrichment of fritillaria from the complex system.

On the basis of traditional MIPs preparation technology, Zhang et al. [56] used hydrogenated quinine as a dummy template molecule and macromolecular aggregation-assisted method to successfully prepare liquid crystalline molecularly imprinted monolith (LC-MIM) for the first time, as shown in Figure 4. A low level of cross-linker (26%) has been found to be sufficient to achieve molecular recognition on the crowding-assisted LC-MIM based on the physical cross-linking of mesogenic groups in place of chemical cross-linking. The satisfied baseline separation of quinidine and quinine could be achieved with good resolution (Rs = 2.96), selectivity factor (α = 2.16) and column efficiency (N = 2650 plates/m).

#### 3.1.3. Terpenoids and Steroids

Terpenes is a generic term for a range of terpenoids, which are olefins with the molecular formula of an integer multiple of isoprene. Terpenes are a group of hydrocarbons of natural origin widely found in plants. Steroids are small, chemically similar lipophilic molecules with a relative molecular mass of around 300 Da. Therefore, they can enter the cell by simple diffusion across the plasma membrane.

Celastrol (Cel), also known as tripterine, is a bioactive component in Tripterygium. In recent years, Cel has attracted wide attention due to its various biological and pharmacological activities, such as anti-inflammatory, anticancer and antioxidant activities. Although Cel has good effects on anticancer and weight loss, Cel has been found to not only cause organ damage of varying degrees to users, but also to lead to fatal acute renal failure through animal and clinical studies in recent years. Therefore, it is very important to accurately control Cel content in Chinese medicinal materials and establish sensitive detection method for Cel in complex samples.

Li et al. [63] proposed a method to prepare MIPs modified with Mn-doped ZnS quantum dots (QDs) as a photosensing material for the specific recognition of celastrol (Cel) in TCM using sol–gel polymerization. The specific preparation steps are shown in Figure 5. Firstly, L-cysteine (L-Cys)-modified Mn-doped ZnS QDs (L-Cys@Mn-ZnS) were used as imprinting substrates. The amino and carboxyl groups on the surface of MN-ZNs QDs can provide more binding sites for polymerization of fluorescent molecularly imprinted polymers. The MIP-functionalized Mn-doped ZnS QDs (MIPs@L-Cys@ Mn-ZNs QDs) showed high selectivity to Cel, and the imprinting coefficient was 14.19. The sensor based on MIPs@L-Cys@Mn-Zns QDs showed a sensitive response to Cel in the linear range of 0.1–3.5 μM, with a LOD of 35.2 nM. MIP-coated QDs have good recovery and reproducibility, which proves that MIP-coated QDs can be used to efficiently and specifically capture, enrich and detect Cel in complex TCM samples.

Chen et al. [65] synthesized novel MMIPs with a specific tripterine recognition ability via precipitation polymerization. The preparation steps are shown in Figure 6. The HPLC-UV results of the analysis showed that the MIPs were specific, rapid and sensitive for the extraction, separation and quantitative determination of rhodopsin. The proposed MMIPs were successfully employed for separating tripterine from crude extracts and real-time sample detection. MIPs may be a practical material for the pretreatment and purification of plant resources and a simple and accurate quantitative analytical method that can provide basic data for in vivo analysis of tripterine and quality control of *T. Wilfordii*.

Li et al. [42] proposed a new method for the preparation of MIP-functionalized magnetic carbon nanotubes (MCNTs) using Cu^2+^-mediated sol–gel polymerization for the selective recognition of Cel in TCM. The preparation steps are shown in Figure 7. The results showed that Cel-MIPs@MCNTs had a fast adsorption kinetic equilibrium time (40 s), a high adsorption capacity (13.35 μg/mg), a satisfactory imprinting factor (IF) of 3.41 and a high magnetic responsivity (44.38 emu/g). The above characterizations indicated that Cel-MIPs@MCNTs are a promising adsorbent for the selective and efficient separation of Cel from complex TCM samples. Combined with HPLC determination, the established method had an excellent linearity ranging 0.15–200 μg/mL with a correlation coefficient of 0.9998, a lower LOD of 0.05 μg/mL and the recoveries of 84.47–91.5% (RSDs < 5.35%) from real samples.

#### 3.1.4. Flavonoids

Quercetin (Qu) is a kind of natural flavanol compound with various biological activities, which plays an important role in antioxidant, tumor inhibition, antibacterial and anti-inflammatory activities, as well as the prevention and treatment of diabetes complications. The concentration of quercetin in natural products is very low, so it is difficult to be separated and purified [78].

Zhu et al. [65] prepared MMIPs of quercetin (Qu/MMIPs) by precipitation polymerization. The results show that Qu/MMIPs has good magnetic properties and specific selectivity of quercetin template molecules, which can be used for the rapid and effective separation of quercetin from complex natural products. Zhang et al. [54] first synthesized a non-toxic dual-template molecularly imprinted polymers (DMIPs) using quercetin and schisandrin b as template molecules and deep-eutectic solvents as functional monomers. The DMIPs were used to efficiently and simultaneously enrich quercetin and schisandrin b from the mixed crude extracts of penthorum and schisandra. The results indicated that the DMIPs exhibited good adsorption kinetics (80 min for adsorption equilibrium) and high selectivity. The largest adsorbing capacities of quercetin and schisandrin b were 23.58 mg/g and 41.64 mg/g, respectively. After presaturation with quercetin and schisandrin b, the nontoxic DMIPs were fed to the mice, in which both templates were successfully detected in blood samples. The experimental results proved that DMIPs can be used to specifically separate and purify target products from some complex biological systems, and that the DMIPs may be used as a potential drug delivery system of compound herbal formulas.

Formononetin has anticancer, antiatherosclerosis, antioxidation and other pharmacological activities. As a flavonoid, formononetin can not only prevent various cancers and inflammation, but also show a wider range of pharmacological effects and low toxicity. Preparative liquid chromatography and column chromatography are mainly used to separate and preconcentrate formononetin. However, these methods have complicated steps and high separation cost, which has directly limited the further application in the medical field. Using formononetin as the template and magnetic Fe_3_O_4_ as the nuclear support material, He et al. [33] used surface imprinting technology and stimulus response imprinting strategy to prepare magnetic thermosensitive molecularly imprinted polymers (MTMIPs) by bulk polymerization. The obtained MTMIPs were used as SPE adsorbent and combined with HPLC to establish a sensitive method for formononetin determination. This method provides a rapid, environmentally friendly and intelligent enrichment method for exploring flavonoids compounds in the complex system of crude extract of red clover.

Calycosin is one of the main components in Radix Astragali. It is an isoflavone compound with various biological activities, such as antioxidation, anti-inflammatory and antitumor activities. The strict control of its content in TCM and dynamic plasma concentrations in vivo have great importance for the efficacy of drugs and the treatment of diseases. Therefore, it is very important to develop a technology for calycosin analysis which is convenient, efficient and exact in the application of the quality control of medicinal herbs and preparations.

Sun et al. [31] constructed a novel molecularly imprinted electrochemical sensor based on nanoporous carbon (NC) and applied it to the sensitive determination of carbenoxol in complex TCM samples. The polymer was bonded to the surface of NC-modified glassy carbon electrode (GCE) by electropolymerization. The electrochemical performance of the proposed sensor (MIPs/NC/GCE) was evaluated by cyclic voltammetry, electrochemical impedance and chronoamperometry. Under optimized conditions, the proposed sensor was used to detect calycosin, showing good reproducibility stability and selectivity. The detection range of the method was 4.20 × 10^−7^–1.29 × 10^−4^ mol/L, and the LOD was 8.5 × 10^−8^ mol/L. Furthermore, this study will provide a prospective exploration of the application of electrochemical analytical techniques in detecting of active components in TCM.

#### 3.1.5. Polyphenols and Organic Acids

Resveratrol is a kind of natural polyphenol substance with abundant biological activity, which mainly exists in TCM materials such as poria cocos and peanut. It is one of the most promising drugs in the treatment of cancer and tumor.

He et al. [37] prepared resveratrol MIPs microspheres by precipitation polymerization as SPE column fillers. The resveratrol from flower roots and stems was separated and purified with a MISPE column. The results showed that the IF of the prepared microspheres was 1.62, and the MISPE column had a significant specific adsorption effect on resveratrol. After separation and purification of resveratrol, the purity of resveratrol could reach 92.5%, and the recovery was 76.2%. The MISPE technology in this study can reduce the amount of reagent used to a certain extent and ensure the purity and recovery of resveratrol, which is of great significance to the efficient and environmentally friendly extraction and purification technology.

Ferulic acid (FA) is the main component of TCM such as *Ferula assafoetida* L., *Angelica sinensis* Diels and *Ligusticum chuanxiong* Hort. FA has been shown to have numerous pharmacological and therapeutic applications including potential treatment for diseases such as cancer, cerebrovascular disease, diabetes and Alzheimer’s disease.

Fu et al. [66] developed a method for the reproducible and sensitive enrichment and determination of FA in Rhizoma Chuanxiong extracts and rat plasma by MISPE-HPLC. The specific preparation steps are shown in Figure 8. The MCNTs@FA-MIPs based on the surface imprinting technology had excellent magnetic properties and a homogeneous appearance. MCNTs@FA-MIPs exhibited good adsorption kinetics (equilibrium time of 2 h), high adsorption capacity (50 mg/g), fast separation, good selectivity and a separation factor of 1.73. The spiked recoveries ranged from 95.53% to 100.03% with RSD < 5.5%. The results showed that MCNTs@FA-MIPs can effectively extract FA from TCM preparations and blood samples with high specificity. In conclusion, the MIPs synthesized in the present study served to enrich and recover FA from biological samples and TCM extracts.

Buffon et al. [67] developed a disposable electrochemical platform, based on a screen-printed electrode modified with rGO, FeNPs and molecularly imprinted PPy film, for vanillic acid (VA) determination. Using optimized conditions, the proposed disposable platform presented linear concentration ranges of 1.0 × 10^−9^ to 1.5 × 10^−7^ mol/L. The limits of detection and quantification obtained for the device were 3.1 × 10^−10^ and 1.0 × 10^−9^ mol/L, respectively. The analytical method developed was successfully applied for VA determination in banana and orange peels, where it presented a good degree of accuracy for the detection of this molecule.

### 3.2. Analysis of Hazadous Components

The residue of hazardous substances in TCM is an important factor that affects the efficiency and restricts the development of TCM. The hazardous substances can be roughly divided into two categories: one is the toxic substances from the product itself; the other is pesticide residues and harmful heavy metal residues. By using MIPs for selective enrichment of targeted hazardous components, accurate analysis of hazardous components can be realized, so as to realize the reduction of drug toxicity [79], the enhancement of effect and improvement of drug safety.

#### 3.2.1. Endogenous Hazardous Substances

Aristolocholic acid is a kind of substance containing nitrophenanthrene carboxylic acid group in aristolochiaceae plants, which is widely used in the treatment of various diseases. Studies have shown that TCM containing aristolocholic acid or derived products are harmful to human body, so it is necessary to monitor the presence of aristolocholic acid in drugs [39].

Zhang et al. [39] used surface imprinting technology via suspension polymerization to prepare the aristolocholic acid core-shell MIPs (SiO_2_@MIPs NPs) and used it as an adsorbent to selectively remove AAI from Caulis Clematidis Armandii. The experimental results show that SiO_2_@MIPs NPs has a high IF (4.9) and a good selectivity for template structure analogues with selection coefficients ranging from 2.3 to 6.6. Finally, SiO_2_@MIPs NPs was used as adsorbent for the pretreatment of standard Caulis Clematidis Armandii sample, with HPLC determination. The recoveries of the method were 73–83%. The results showed that SiO_2_@MIPs NPs could be used as a highly selective material for the selective separation and analysis of aristolochic acid in TCM. Lu et al. [49] prepared AAI MIPs by forming a surface-imprinted layer on the Fe_3_O_4_@SiO_2_. The MIPs obtained in this study provide new ideas about the optimization of pretreatment steps for the separation, enrichment and detection of AAI in TCM.

Li et al. [68] prepared MIP-functionalized MCNTs via a typical sol–gel copolymerization procedure for the effective removal of AAΙ from TCM. The MCNTs@AAI-MIPs were prepared using a facile and environmentally friendly sol–gel method. The results showed that the imprinted nanocomposites had fast separation speed (10 s), high adsorption capacity of 18.54 g/mg, short kinetic equilibrium time (15 min) and excellent selectivity with the IF of 3.17. The LOD (S/N = 3) was 0.034 μg/mg, and the recoveries ranged from 80–110% (RSD = 3.27–8.16%) in the spiked TCM samples. The results indicated that the proposed MCNTs@AAI-MIPs can efficiently and specifically capture AAI from practically complex TCM samples.

Aconitine (ACO) is a diterpene alkaloid derived from some TCM products, such as Aconitum carmichaeli Debx. It plays an important role in pathological activities, such as cardiovascular diseases and rheumatic diseases. However, the pharmacodynamic effect of ACO is dose-dependent; the low dose of ACO can protect the cardiac muscle cells, but the high dose can cause cardiac toxicity. Therefore, the development of a valid and sensitive method to accurately determine the content of ACO in the drugs containing ACO is of great importance.

Luo et al. [57] proposed a novel molecularly imprinted ratio fluorescence sensor (MI-RFL sensor) method by precipitation polymerization using ACO as the template, blue emitting FeS_2_ QDs as the fluorescence sensor and fluorescence response signal material and yellow emitting FeS_2_ QDs as the reference signal materials. The sensor can be used to detect acetylcholine in TCM with high selectivity and sensitivity. Under the optimal conditions, the fluorescence intensity ratio (I_443_/I_590_) of the MI-RFL sensor had a good linear relationship with ACO concentration in the range of 0.05–5.0 μM, and the LOD was 24 nM. The established method was applied to the determination of acetylcholine in Fuzi Lizhong pills with satisfactory results. It provides a reference for the application of FeS_2_ multicolor emission QDs and the detection of harmful alkaloids.

#### 3.2.2. Pesticide Residues

Ginseng has long been used as TCM in China. Ginseng saponin is main composition of ginseng, which can enhance human immunity, protect cardiovascular system, protect the central nervous system, regulate endocrine, preventing diabetes, and prevent tumors and cancer. Organophosphorus, organochlorine and carbamate pesticides are heavily used throughout the growing cycle of ginseng. The use of these pesticides will make ginseng have a large number of pesticide residues. Therefore, it is urgent to use scientific method to analyze pesticide residues in ginseng and establish the maximum residue limits in ginseng. At present, according to the national standard GB2763-2016, only two pesticides (fenoxazole and pyrimethoril) were registered in ginseng with the maximum residue limits (MRL) food. Pyraclostrobin is a kind of methoxyacrylate fungicide.

Wang et al. [69] established a method of molecularly imprinted dispersed solid-phase extraction (MIDSPE) combining with HPLC to selectively preconcentrate and detect the content of pyraclostrobin in ginseng samples. The recoveries were 77.60–93.15%, and the RSD values were 2.47–6.99% (*n* = 5). Compared with C_18_ as adsorbent, the recovery rate of pyraclostrobin was higher. This analysis method can achieve good recovery, repeatability and reproducibility. The LOD (S/N = 3) was 0.01 mg/kg. In this study, the MIDSPE -HPLC method was used to demonstrate the specific selection and separation of trace zazamide in ginseng samples successfully. The results showed that the residual amount of pyraclostrobin in ginseng was lower than the MRL of 0.1 mg/kg in the European Union. In the actual ginseng sample residue detection experiment, researchers purchased fresh ginseng from five producing areas (Fusong, Changbai, Yanji, Ji’an and Liaoning Hengren), and the results showed that pyraclostrobin was detected in ginseng from Fusong, Changbai and Yanji Hengren. The residues were lower than the European MRL of 0.1 mg/kg. This method proposed a new possibility for the determination of pesticide residues in ginseng.

Cyfluthrin is a widely used insecticide. Hu et al. [50] prepared MMIPs by precipitation polymerization to extract and enrich cyfluthrin, which was used to analyze pesticide residues in TCM. The polymer successfully adsorbed cyfluthrin from honeysuckle using unshaped Fe_3_O_4_ particles as magnetic core and was quantitatively detected by HPLC. Under optimized conditions, the LOD and limit of quantification (LOQ) were 32.987 ng/mL and 109.955 ng/mL, respectively. The recoveries of real samples were 91.5–97.2% with RSD of 5.35–8.32%. The MMIPs is easy to separate and prepare, so this method has great potential in the analysis of cyfluthrin in complex matrix such as TCM.

Wu et al. [70] prepared a molecularly imprinted membrane by precipitation polymerization combining with electrospray mass spectrometry (MIM-ESI MS) to rapidly detect four pesticide residues in TCM, including organophosphorus (OPP), carbamate, pyrethroids and isoniazid. Due to the specific recognition and enrichment of analytes, MIM-ESI can reduce the LOD of traditional analytical methods by a factor of 50 compared to our previous environmental ionization method, MESI. The sensitivity and specificity of the method were further verified. The data showed that MIM-ESI MS has satisfactory performance and can directly and rapidly quantify different pesticide residues in TCM samples. It is suitable for the direct quantitative analysis of pesticide residues in TCM. This testing technique may help ensure the quality of TCM in the future as well.

Fatah et al. [71] proposed a sensitive, simple, low-cost and highly selective Metribuzin (MTZ) electrochemical sensor based on MIT by self-assembly bulk polymerization approach. The electrochemical sensor was designed for its detection in commercial product (Egyscor^®^ 70%) (Bharat Rasayan Limited Company, India), spiked tomatoes and potatoes samples with recovery values ranging from 97.12 to 103.41%. The sensor showed selective adsorption ability and a good linearity over the concentration range of 0.2 ng/mL to 21.429 µg/mL, with LOD and LOQ of 0.1 pg/mL and 0.3 pg/mL, respectively. All of the above indicate the efficiency and promising applicability of the proposed sensor in regulatory food agencies in order to have fast decisions regarding contamination of agricultural samples, if any, with MTZ.

#### 3.2.3. Heavy Metal Pollution

Heavy metal ions are a class of pollutants with a wide distribution range and high toxicity. The accumulation of a certain amount of heavy metals in the human body can cause chronic poisoning. Heavy metals are very difficult to biodegrade; on the contrary, they can be enriched thousands of times under the biomagnification of the food chain, and finally enter the human body. Heavy metals in the human body can have a strong interaction with proteins and enzymes and may also accumulate in some organs of the human body, resulting in chronic poisoning. In addition, heavy metals such as mercury, lead and chromium can also cause environmental pollution. Therefore, it is necessary to detect them effectively in TCM.

Dadfarnia et al. [72] synthesized vanadium (V) ion-imprinted polymers with V as the template and N-toluene-N-phenylhydroxylamine as the functional monomer. Using ion-imprinted polymers as adsorbents, a solid-phase extraction method was developed to adsorb V, combined with electrothermal atomic absorption spectroscopy determination. The maximum adsorption capacity for V ions at pH = 4.0 is 26.7 mg/g. Under optimal conditions, the enrichment coefficient was 289.0 and the LOD was 6.4 ng/L. The established method was successfully applied to the determination of vanadium in parsley, zucchini, black tea, rice and water samples.

#### 3.2.4. Additional Illegal Drugs

Asgharinezhad et al. [73] prepared polypyrrole-polyaniline nanorods (PPy-PANI NRs) via oxidative polymerization and they subsequently were used in the electromembrane extraction of phenolphthalein from herbal slimming products. This is the first report on the application of PPy-PANI NRs as an additive in the electromembrane extraction method. Finally, the new method was successfully used to determine PP in slimming products of Iranian markets, and satisfactory results (5.5–8.5% RSDs; relative recovery, 80–110%) were obtained. The new strategy is quick, easy and cheap, and it is environmental-friendly owing to the few µL use of the organic solvent. A wide linear range and low detection limit equal to 3.0–2000 and 1.0 μg/L were obtained, respectively. Besides, the method needs a low sample volume (8.0 mL) for each batch and represents excellent clean-up compared to the more sample preparation methods.

### 3.3. Other Applications of MIT in TCM

#### 3.3.1. New Formulation of TCM 

MIPs hydrogels have good biocompatibility and have great application prospects in sustained release preparations. Gao et al. [40] used MIT to transform the drug dosage form of oral huperzine A. By using hydroxypropyl methyl cellulose as the raw material and using reverse suspension polymerization, effluent gel microspheres were prepared as a drug delivery platform. The results showed that molecularly imprinted water-soluble microspheres sustained a release effect on huperzine A. It provides the experimental basis for the modification of MIPs microspheres with new dosage forms.

#### 3.3.2. Chiral Drug Resolution

Chirality is a ubiquitous phenomenon in nature. The enantiomers of a chiral molecule despite the similar physical and chemical properties exhibit distinct pharmacological, biological and toxicological activities. An ideal chiral discrimination strategy can be realized by absolute enantiomeric discrimination in racemates, high recognition sensitivity and enforceability for wide varieties of chiral molecules regardless of molecular properties [80]. Importantly, MIT-combined surface-enhanced Raman scattering (MIT-SERS) have been used for high-throughput precise chiral identification of amino acid analytes [80]. 

For TCM, Balamurugan et al. [74] developed a HPLC column with strong separation ability for two kinds of chiral ephedrine. Through using two types of ephedrine molecules with different optical activity as template molecules, they prepared MIPs by non-covalent molecular imprinting, and found that this MIPs filler was effective for separation of ephedrine from racemes. It provides technical support for the application of MIPs in the chiral resolution of drugs and creates favorable conditions for rational drug use and drug safety.

#### 3.3.3. Detection of Growing Environment

The detection of the growing environment is important to assess TCM and ensure quality. Cheng et al. [75] used surface imprinting strategy to prepare surface-imprinted MIPs and used them as a new SPE filler. After several tests on water samples, it was confirmed that this MIPs sorbent was superior to commercially available SPE adsorbents. Therefore, the MIPs can be applied in MISPE column with HPLC determination to analyze sulfanilamide armourazole in soil and water. Li et al. [76] used electrochemical in situ polymerization to prepare MIPs with the pesticide carbazim (CBD) as the template molecule, and they combined MIPs with ZIF67@Ni to construct a new electrochemical sensor, which can analyze and detect CBD in water and soil with high selection, stability and repetition. This work can inspire scientists to employ MOF and MIP as electrochemical sensing agents, which may play an important role in drug monitoring and drug quality control.

## 4. Conclusions and Prospects

In summary, a variety of enrichment methods developed based on MIPs can complete the selective extraction of a variety of natural active components in TCM. Thanks to the above efficient extraction methods, traditional analysis instruments like sensors and liquid-phase chromatography can sensitively quantitatively analyze the enriched products, which solves the problem of analyzing active components from TCM. The detection methods of heavy metals, pesticides and other hazardous components have been developed, which effectively ensures the safety of the use of TCM in the future. Moreover, the promotion of MIPs for the study of TCM metabolism should not be underestimated. Related research studies have displayed the advantages of high selectivity, high sensitivity, rapid detection and simple synthesis and strong operability. 

Several prospects can be expected as follows. 

(1)The enrichment effect of MIPs on specific components of TCM is very obvious, which can be used as an important means of targets enrichment of TCM. Traditional separation methods (solvent extraction, paper chromatography, common column chromatography, etc.) and MIPs adsorption materials are effectively complementary. The former is responsible for crude separation of samples, and the latter is responsible for fine separation that can improve the efficiency and quality of enrichment and separation work.(2)For precious TCM products, only a small number of samples can be taken to realize the quality analysis of medicinal materials. Therefore, using MIPs as adsorbents not only ensures that precious TCM products are not wasted, but also realizes the quality detection of precious TCM products, which is of great significance for the quality control and safe drug use of precious TCM products.(3)MIPs are rarely used in TCM preparations, and the application of stimulus response strategy should be strengthened in TCM slow and controlled release preparations, separation and purification and formulation typing. The application of such a pH-responsive nano-carrier might offer a potential platform for controlled delivery and increasing the bioavailability of drugs.(4)The MIPs’ function of selective recognition and targeted delivery/localization leads to perspective research, namely, the early detection and treatment of related diseases using MIT in TCM.(5)MIPs bind to target molecules rapidly; therefore, the combination with fluorescence sensing technology can greatly shorten the quality inspection time of relevant samples. If the corresponding detection kit based on MIPs is developed, it is promising to realize real-time field detection of the growth environment of TCM and improve the detection efficiency.(6)MIPs are simple to prepare and easy to operate. If the laboratory preparation process can be successfully extended to industrial production, it will provide favorable conditions for its wider use. Since most of these technologies are still confined to the lab, pilot-scale investigations and tests are necessary to ensure their reproducibility and scalability in the real world.(7)Green aspects of MIPs in TCM research should be considered. The environmental consequences of unsustainable MIT provide adequate impetus for change towards GREENIFICATION. The progress that brought about the greenification of MIT is mainly derived from two aspects: operator health risks are greatly reduced, and the associated adverse effects on the environment are minimized [81,82].

In addition, MIPs can be used for quality control of various proprietary TCM preparations with their strong enrichment ability for special components, such as detecting whether the content of active components in the main drug reaches the standard. Taking advantage of the slow and controlled release capability of MIPs, various dosage forms of TCM can be prepared, such as sustained-release microspheres, transdermal patches and ointment. In short, MIT will definitely promote the development of TCM research in the future.

## Figures and Tables

**Figure 1 molecules-28-00301-f001:**
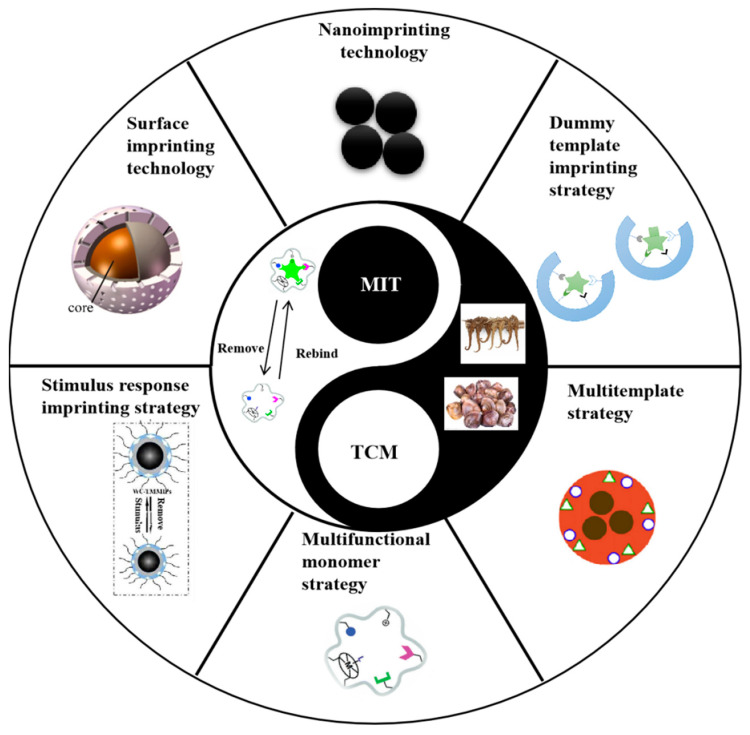
New imprinting technologies and strategies for preparing TCM-related MIPs [23,43].

**Figure 2 molecules-28-00301-f002:**
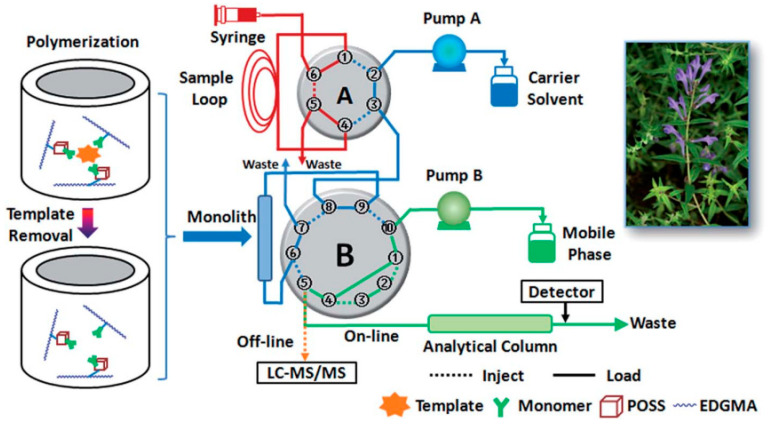
Scheme of the preparation and application of in-tube SPME-HPLC [58].

**Figure 3 molecules-28-00301-f003:**
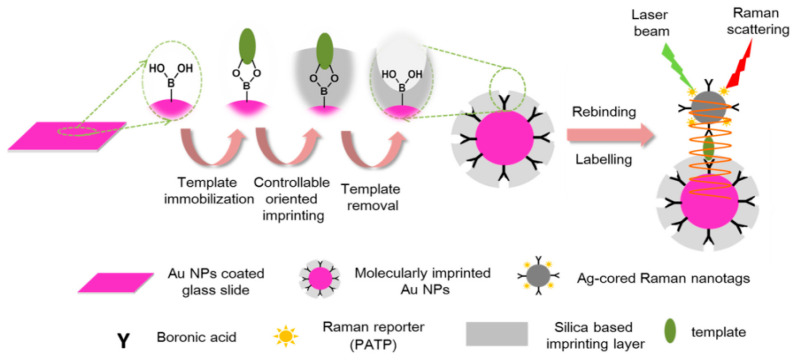
Preparation method and principle of plasma immuno-sandwich assay of bis-borate affinity nanoparticles [48].

**Figure 4 molecules-28-00301-f004:**
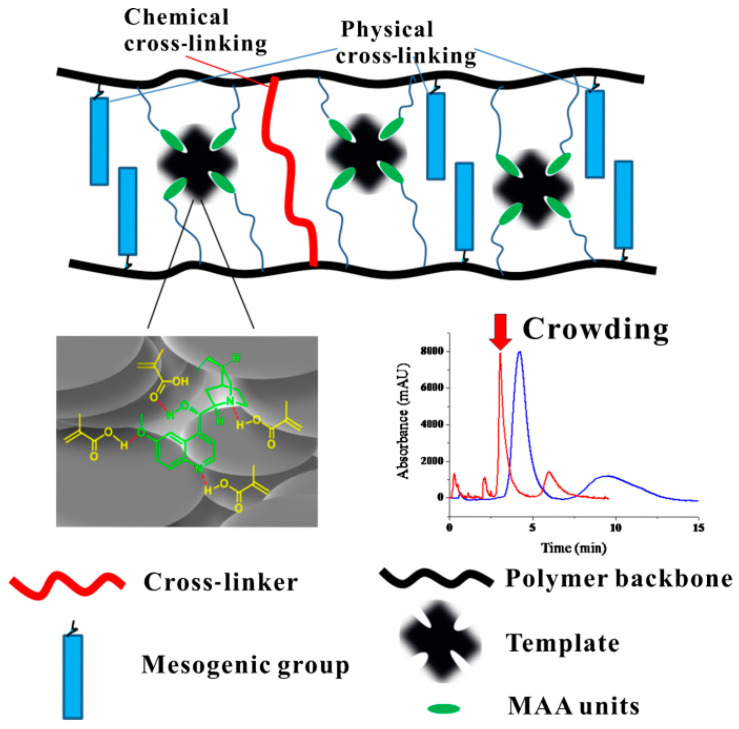
Schematic diagram of LC-MIM preparation process [56].

**Figure 5 molecules-28-00301-f005:**
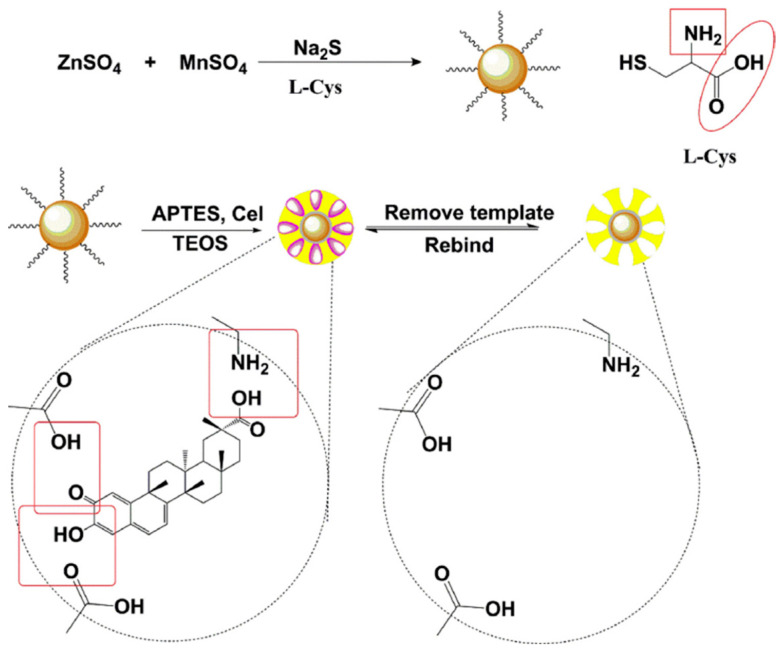
Schematic illustration of preparation procedure of MIPs@L-Cys@Mn-ZnS QDs [63].

**Figure 6 molecules-28-00301-f006:**
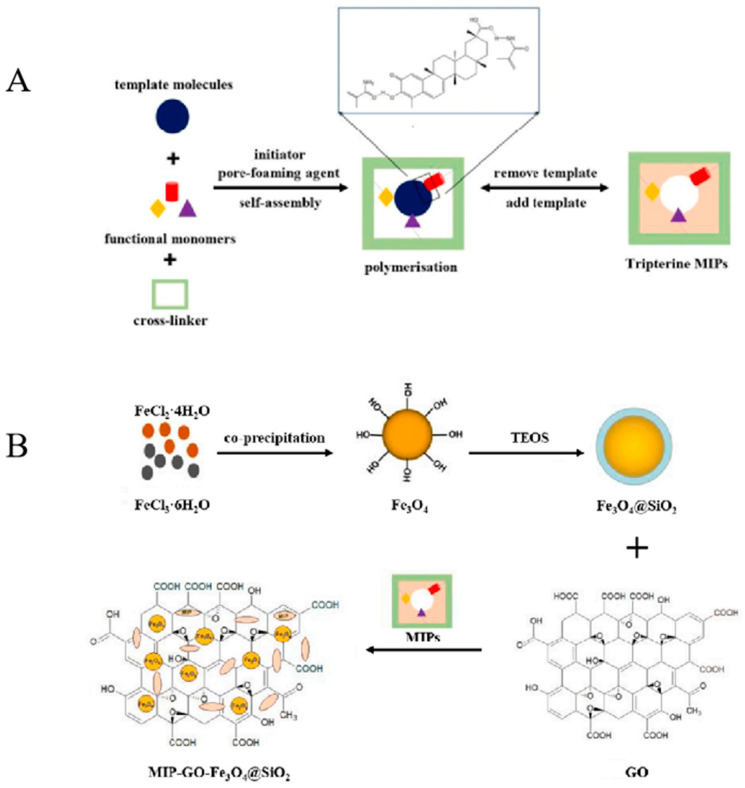
(**A**) The preparation and application process of MIPs. (**B**) The preparation process of magnetic graphene oxide molecularly imprinted polymers (MIPs-GO–Fe_3_O_4_@SiO_2_) [65].

**Figure 7 molecules-28-00301-f007:**
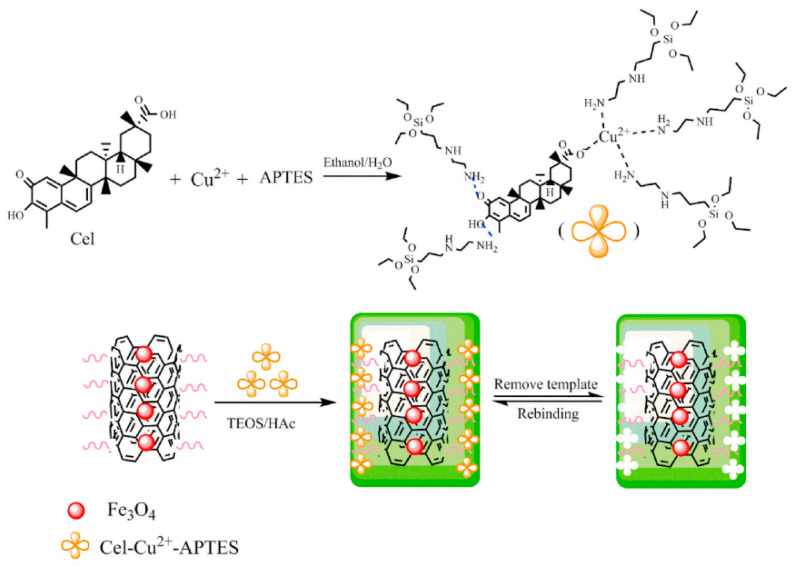
The synthesis procedure of Cel-MIPs@MCNTs [42].

**Figure 8 molecules-28-00301-f008:**
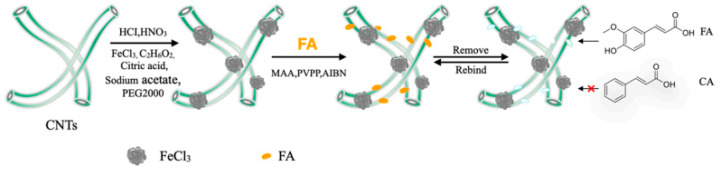
The synthesis procedure of MCNTs@FA-MIPs [66].

**Table 1 molecules-28-00301-t001:** Comparison of several separation methods for TCM.

Separation Methods	Advantages	Disadvantages	Ref.
extraction separation	The instrument is simple and easy to operate, with a wide range of application.	It takes a lot of work to analyze batch samples, and the extraction solvent is often volatile, flammable and has certain toxicity.	[7]
chromatographic separation	Both the required equipment and the operation procedure are simple. It shows good separation effect, high repeatability, relaxed separation condition and wide application range.	The resolution is not high, and the separation operation is slow.	[9,10]
recrystallization	The separation equipment requirements are low, the operation is simple and the solvent amount is less.	It produces bubbles that can induce nucleation, difficult to predict seed point and unmeasurable scale-up production.	[14,23]
membrane separation	It has good flexibility, strong operability, high selectivity, less membrane fouling and easy scalability.	The product cannot be condensed into dry matter and the isomers cannot be separated.	[11]
molecular distillation	It just needs low operating temperature and can greatly save energy consumption. The pressure of distillation is low. The separation efficiency is high.	A small amount of vaporization occurs and production capacity is not big. It needs auxiliary system and high vacuum operation leading to high maintenance cost.	[12]
macroporous resin separation	It shows selective adsorption of pH-sensitive active components, easy elution, decolorization and deodorization, high stability and reusability.	The requirements for technical conditions are stricter, the operation is more complex and the requirements for pre-processing are strict.	[13]
high performance liquid chromatography (HPLC)	It is simple and sensitive.	The equipment is expensive, time consuming and solvent consuming	[24,25]
MIT	It shows high affinity and selectivity, strong ability to resist harsh environment, good stability, simple preparation and low cost.	It shows poor penetration and less commercialization.	[14]

**Table 2 molecules-28-00301-t002:** Applications of MIT in TCM.

Applications	Classification of Active Components	Analytes	Imprinting Technology/Strategy	Polymerization Method	LODs	Detection Technique	Real Sample	Recovery/%	Ref.
Active Components	glycosides	baicalin	-	-	0.001 mg/mL	LC-MS/MS	scutellaria baicalensis	-	[58]
		hesperidin	-	electro-polymerization	1.4 μΜ	electrochemical detection	chenpi	99.00–104.40	[59]
		salidroside	-	bulk polymerization	0.21 μg/L	HPLC-UV	rhodiola crenulata root	88.74–97.64	[60]
		ginsenosides	surface imprinting technology	-	1.7 ng/mL	HPLC-UV	ginseng	-	[48]
		ginsenoside Rg_1_	surface imprinting technology	precipitation polymerization	-	HPLC-UV	-	-	[35]
	alkaloids	caffeine	stimulus response imprinting strategy	coprecipitation	-	HPLC-UV	-	-	[61]
		labaaconitine	-	bulk polymerization	-	HPLC-UV	-	-	[32]
		matrine alkaloids	-	precipitation polymerization	-	HPLC-MS/MS	bean fruit extract	89.6–98.0	[36]
		peimine	-	-	2.0 × 10^−7^ mol/L	UV	Unibract Fritillary Bulb	97.2–102.0	[62]
		cinchona alkaloids	dummy template imprinting	-	-	HPLC-UV	-	-	[56]
	terpenoids and steroids	celastrol	surface imprinting strategy	sol–gel polymerization	35.2 nM	MIR sensor	triptergium wilfordii hook F	88.0–105.0	[63]
		tripterine	stimulus response imprinting strategy	-	-	HPLC-UV	crude extract of t. wilfordii,	-	[64]
		celastrol	multifunctional monomer strategy	sol–gel polymerization	0.05 μg/mL	HPLC-UV	TCM samples	84.47–91.5	[42]
	flavonoids	quercetin	stimulus response imprinting strategy	precipitation polymerization	-	HPLC-UV	-	-	[65]
		quercetin and schisandrin B	multitemplate imprinting strategy	bulk polymerization	-	HPLC-UV	mice	-	[54]
		formononetin	multifunctional monomer strategy, surface imprinting technology, stimulus response imprinting strategy	-	0.017 μg/mL	HPLC-UV	daidzein, formononetin, genistein	-	[33]
		calycosin	-	electro-polymerization	8.5 × 10^−8^ mol/L	electrochemical sensor	radix astragali	99.6–100.4	[31]
	polyphenols and organic acids	resveratrol	-	precipitation polymerization	-	HPLC-UV	sea water	76.2	[37]
		ferulic acid	stimulus response imprinting strategy	precipitation polymerization		HPLC-UV	ligusticum chuanxiong extracts and rat plasma	98.65–110.03	[66]
		vanillic acid (VA)	-	electro-polymerization	3.1 × 10^−10^ mol/L	voltammetric detection	banana and orange peels	-	[67]
Hazadous Components	endogenous hazadous substances	aristolochic acids	surface imprinting technology	-	0.033 μg/mL	HPLC-UV	Kebia trifoliate	73–83	[39]
		aristolochic acids	surface imprinting technology and stimulus response imprinting strategy	-	-	HPLC-UV	-	-	[49]
		aristolochic acid	stimulus response imprinting strategy, surface imprinting strategy	sol–gel polymerization	0.034 ug/mg	HPLC-UV	TCM	80–110	[68]
		aconitine	-	precipitation polymerization	24 nM	MIR sensor	fuzi lizhong pills	95.2–103.1	[57]
	pesticide residues	pyraclostrobin	-	-	0.01 mg/kg	HPLC-UV	ginseng	77.60–93.15	[69]
		cyfluthrin	surface imprinting technology and stimulus response imprinting strategy	-	32.987 ng/mL	HPLC-UV	honeysuckle	91.5–97.2	[50]
		organophosphorus (OPP), carbamates, pyrethroids and neonicotinoids	-	-	0.5 ng/mL 1.0 ng/mL 0.5 ng/mL 0.1 ng/mL	MS	panax notoginseng, angelica sinensis, codonopsis pilosula	95.1–102.2	[70]
		metribuzin		bulk polymerization	5.00 × 10^−10^ M (0.1 pg/mL)	MIP-carbon-paste sensor	tomatoes and potatoes	97.12–103.41	[71]
	heavy metal pollution	V(V) ions	-	bulk polymerization	6.4 ng/L	ETAAS	parsley, zucchini, black tea, rice, and water samples	about 96	[72]
	additional illegal drugs	phenolphthalein	-	oxidative polymerization	1.0 μg/L	LC-UV	herbal slimming products	80–110	[73]
other applications	new formulation of TCM	huperzine A	-	suspension polymerization	0.37 μg/L	HPLC-UV	rat	93.82–94.48	[40]
	chiral drug resolution	ephedrine enantiomers	-	bulk polymerization	-	LC-UV	-	above 92	[74]
	detection of growing environment	sulfamethoxazole	surface imprinting technology	-	-	HPLC-UV	river water, rainwater, soil, sediment, pork, egg	91–106	[75]
		carbendazim	-	electro-polymerization	1.35 × 10^−13^ M	electrochemical sensor	soil and water	above 98	[76]

Note: “-” indicates “none”; LODs: limit of detections; HPLC: high-performance liquid chromatography; HPLC-UV: high-performance liquid chromatography-ultraviolet spectrometry; LC-MS/MS: liquid chromatograph mass spectrometry/mass spectrometry; MIR sensor: molecular imprinting ratiometric fluorescence sensor; MS: mass spectrometry; ETAAS: electrothermal atomic absorption spectrometry; LC-UV: liquid chromatography–ultraviolet spectrometry; LC: liquid chromatography.

## Data Availability

Not applicable.

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
