# Peer review of "Applications of Molecular Imprinting Technology in the Study of Traditional Chinese Medicine"

_molecules, 2022, doi:10.3390/molecules28010301_

Round 1
Reviewer 1 Report
The manuscript ‘Applications of Molecular Imprinting Technology in the Study of Traditional Chinese Medicine’ disclosing molecular imprinting technology as an interdisciplinary technology for medicine industry. The manuscript focuses on an interesting and hot topic; however, it is still very immature. The list of the corrected points described below, especially the frame structure of MS should be reconsidered. The language needs polishing.
LINE 20 The abstract of this article is a bit weak and the details should be provided.
LINE 45-46 Please provide the citation.
LINE 46-55 Please provide the citation.
LINE 55-58 Please provide the citation.
LINE 69-71 List a TABLE to explain the the advantages and disadvantages comparing MIT. Please provide the citation.
LINE 82-83 Why review the advances from 2017 to now, rather than earlier? There seems to be no specific explanation. Introduce the development of MIT in TCM briefly and provide the citation.
LINE 90-205 More cases in TCM should be supplement and summarized. For broad readership, it is very important to cite and provide more international information.
LINE 206-601 This passage is just a simple list without comment. The authors should summarize and review according to the literatures.
LINE 626 Quality control and the early detection and treatment using MIT in TCM should be summarized as a separate heading and paragraph.
Academic Editor Notes
(I) Please check that all references are relevant to the contents of the manuscript.
(II) Any revisions to the manuscript should be marked up using the “Track Changes” function if you are using MS Word/LaTeX, such that any changes can be easily viewed by the editors and reviewers.
(III) Please provide a cover letter to explain, point by point, the details of the revisions to the manuscript and your responses to the referees’ comments.
(IV) If you found it impossible to address certain comments in the review reports, please include an explanation in your appeal.
(V) The revised version will be sent to the editors and reviewers.
Response: We highly appreciate the important advice. We have carefully considered all comments, especially the advised five points, and accordingly revised the manuscript. We sincerely wish a significant improvement can reach the Editor’s requirements and the Journal’s standard.
Detailed Responses to Comments on molecules-2065256
We highly appreciate the reviewers’ valuable comments, significant guidance and sincere endeavors. We have taken into careful consideration of all the comments and substantially revised the manuscript accordingly. And point-by-point responses to the comments have been made as follows. The main revisions have been given, which are marked in RED words in the revised edition. The total manuscript has been carefully checked and polished for writing-quality improvement and presentation clarity.
Reviewer 1:
Open Review
(x) I would not like to sign my review report
( ) I would like to sign my review report
English language and style
( ) English very difficult to understand/incomprehensible
(x) Extensive editing of English language and style required
( ) Moderate English changes required
( ) English language and style are fine/minor spell check required
( ) I don't feel qualified to judge about the English language and style
Response: We highly appreciate the important advice. We have carefully made extensive editing of English language and style.
Comments and Suggestions for Authors
The manuscript ‘Applications of Molecular Imprinting Technology in the Study of Traditional Chinese Medicine’ disclosing molecular imprinting technology as an interdisciplinary technology for medicine industry. The manuscript focuses on an interesting and hot topic; however, it is still very immature. The list of the corrected points described below, especially the frame structure of MS should be reconsidered. The language needs polishing.
Response: We highly appreciate the reviewer’s valuable comments, positive approval and sincere contributions. We have carefully addressed the important concerns and accordingly revised the manuscript for improvement. We quite agree and adjusted the frame structure. The language has also been polished.
1.LINE 20 The abstract of this article is a bit weak and the details should be provided.
Response: Thanks very much for the important guidance and kind reminder. We have provided the details in the abstract. As follows, “…and identification analysis of hazardous components. Fundamentals of MIT are briefly outlined and emerging preparation techniques for MIPs applied in TCM are highlighted, such as surface imprinting, nanoimprinting, multitemplate and multifunctional monomer imprinting. Then, applications of MIPs in common active components researches including flavonoids, alkaloids, terpenoids, glycosides and polyphenols etc. are respectively summarized, followed by screening and enantioseparation. Related identification detection of hazardous components from TCM itself, illegal addition, or pollution residues (e.g., heavy metals, pesticides) are discussed. Moreover, the applications of MIT in new formulation of TCM, chiral drug resolution and detection of growing environment are summarized.” (Line 30-38).
2.LINE 45-46 Please provide the citation.
Response: Thanks very much for the valuable suggestion. We now add the citation in the revised manuscript. As follows, “Especially after Youyou Tu from China won the Nobel Prize for her immense contributions to artemisinin, the research enthusiasm of TCM has been greatly aroused globally [2].” (Line 49).
[2] Wang, J.; Xu, C.; Wong, Y.; Li, Y.; Liao, F.; Jiang, T.; Tu, Y. Artemisinin, the magic drug discovered from traditional Chinese medicine. Engineering 2019, 5, 32-39.
3.LINE 46-55 Please provide the citation.
Response: We provide the citation. As follows, “(4) according to natural properties and affinities [4].” (Line 54).
- Shang, Y.; Xu, X. Advance of experimental studies on protective effect of traditional Chinese medicines and their extracts on cerebral ischemia (in Chinese). Chin. J. Chin. Mater. Med. 2013, 38, 1109-1115.
4.LINE 55-58 Please provide the citation.
Response: We have provided the citation. As follows, “TCM works through the various physiologically active chemicals that contains. Most of the chemical active components of TCM are characterized by low contents, complex structures, diverse types and unstable properties [5].” (Line 60).
[5] He, M.; Ye, Y.; Liu, Y.; Li, Z.; Jiao, B.; Zeng, S.; Su, X. Molecularly imprinted solid-phase extraction combined with high performance liquid chromatography for separation and enrichment of four flavanones in citrus aurantium (in Chinese). J. Instrumental Anal. 2017, 36, 325-330.
- LINE 69-71 List a TABLE to explain the the advantages and disadvantages comparing MIT. Please provide the citation.
Response: Table 1 now added for the comparison of several molecular imprinting polymerization methods for TCM related MIPs preparation.
Table 1 Comparison of several separation methods for TCM [13, 22-23, 26].
|
separation methods |
advantages |
disadvantages |
Ref. |
|
HPLC |
simple and sensitive |
The equipment is expensive, time consuming and solvent consuming. |
[22-23] |
|
repeated recrystallization operations |
The separation equipment requirements are low, the operation is simple, the solvent amount is less. |
It produces bubbles that can induce nucleation, difficult to predict seed point and unmeasurable scale-up production. |
[13,22] |
|
MIT |
high affinity and selectivity, strong ability to resist harsh environment, good stability, simple preparation, low cost |
poor penetration, less productization |
[26] |
6.LINE 82-83 Why review the advances from 2017 to now, rather than earlier? There seems to be no specific explanation. Introduce the development of MIT in TCM briefly and provide the citation.
Response: According to the literature review, there are few related reports on MIT for TCM study before 2017. Most reports have been increasingly published since 2017. Related descriptions are now added for explanation, as follows, “Therefore, in this work, the applications of MIT in the study of TCM are reviewed comprehensively, focusing on the recent advances since 2017 considering a large number of reports are published from 2017.”.
7.LINE 90-205 More cases in TCM should be supplement and summarized. For broad readership, it is very important to cite and provide more international information.
Response: Thanks very much for the important guidance. As guided, we have supplemented more cases ( and added the international example in section “3. Applications of MIPs in TCM Study”. For example, Line 176-178, Line 189-191, Line 200-205, Line 208-216, Line 224-228, Line 236-242.
Fatah et al. [68] proposed ...with MTZ. (Line 569-578).
Buffon et al. [73]developed...for vanillic acid (VA) determination. ....where it presented a good degree of accuracy for the detection of this molecule. (Line 677-686).
3.2.4. Additional Illegal Drugs
Asgharinezhad et al. [75] prepared polypyrrole-polyaniline nanorods (PPy-PANI NRs) via oxidative polymerization and subsequently ... relative recovery, 80-110 %) were obtained. ...more sample preparation methods. (Line 706-716).
8.LINE 206-601 This passage is just a simple list without comment. The authors should summarize and review according to the literatures.
Response: Thanks very much for the valuable suggestion and guidance. As guided, related comments are now added in the revised manuscript, for example, Line 312-313, Line 494-497, Line 563-565, Line 752-754.
9.LINE 626 Quality control and the early detection and treatment using MIT in TCM should be summarized as a separate heading and paragraph.
Response: We highly appreciate the important guidance. About TCM quality control, in section “3.1. Purification Enrichment and Determination of Active Components” and “3.2. Analysis of Hazadous Components” are discussed The contents of the Components are the quality control contents of TCM. Related descriptions are now added to emphasize in “3. Applications of MIPs in TCM Study”. For example, “Nowadays, TCM has been remarkably increasingly used to treat related diseases in clinical practice, ... The two major applications are the purification enrichment and determination of active components and analysis of hazardous components. Thus the quality of TCM can be controlled.” (Line 252-253)
About “the early detection and treatment using MIT in TCM”, I think you mean disease early detection and treatment by using TCM based MIT. Related researches are not found, but as an important perspective, related contents are now added in Section “Conclusions and Prospects”, as follows, “4) The MIPs’ function of selective recognition and targeted delivery/localization leads to perspective researches, namely, the early detection and treatment of related diseases using MIT in TCM.” (Line 784-786)

Reviewer 2 Report
In this work, the authors comprehensively reviewed the recent advances of MIT for TCM studies since 2017, focusing on two main aspects including extraction/separation and purification and detection of active components, and identification analysis of hazardous components. The work is interesting, and which can be published after a minor reversion by replying the following questions:
1. In Introduction section: the authors should address previous
Reviews involving MIT and TCM. Why is there still an empty space for you to write such a review?
2. In Section 2: I suggest the authors to provide a schematic diagram to clear describe the principle of each method.
3. In Sections 2 and 3: A table for summary of discussions might be better.
4. Emulsion polymerization is mentioned in Section 2, but there is no corresponding section in the following text.
5. The classification of different methods in Section 2 must be reconsidered. I suggest: a. Polymerization Methods and New Imprinting Technologies and Strategies should be reviewed as two sections. And b. In New Imprinting Technologies and Strategies, I suggest the authors should be discuss in the following way: New imprinting methods (nano imprinting and surface imprinting); new template strategies (dummy temple and multiple temple); new monomer strategies (multifunctional monomer and stimulus response).
6. When describing the methods, please address the disadvantages and advantages of each one.
7. Please add more your own critical ideas in the text.
Academic Editor Notes
(I) Please check that all references are relevant to the contents of the manuscript.
(II) Any revisions to the manuscript should be marked up using the “Track Changes” function if you are using MS Word/LaTeX, such that any changes can be easily viewed by the editors and reviewers.
(III) Please provide a cover letter to explain, point by point, the details of the revisions to the manuscript and your responses to the referees’ comments.
(IV) If you found it impossible to address certain comments in the review reports, please include an explanation in your appeal.
(V) The revised version will be sent to the editors and reviewers.
Response: We highly appreciate the important advice. We have carefully considered all comments, especially the advised five points, and accordingly revised the manuscript. We sincerely wish a significant improvement can reach the Editor’s requirements and the Journal’s standard.
Detailed Responses to Comments on molecules-2065256
We highly appreciate the reviewers’ valuable comments, significant guidance and sincere endeavors. We have taken into careful consideration of all the comments and substantially revised the manuscript accordingly. And point-by-point responses to the comments have been made as follows. The main revisions have been given, which are marked in RED words in the revised edition. The total manuscript has been carefully checked and polished for writing-quality improvement and presentation clarity.
Reviewer 2
Open Review
(x) I would not like to sign my review report
( ) I would like to sign my review report
English language and style
( ) English very difficult to understand/incomprehensible
( ) Extensive editing of English language and style required
( ) Moderate English changes required
(x) English language and style are fine/minor spell check required
( ) I don't feel qualified to judge about the English language and style
Response: We highly appreciate the reviewer’s valuable comments, positive approval and sincere contributions. We have carefully clarified the important points, and accordingly revised the manuscript for significant improvement.
Comments and Suggestions for Authors
In this work, the authors comprehensively reviewed the recent advances of MIT for TCM studies since 2017, focusing on two main aspects including extraction/separation and purification and detection of active components, and identification analysis of hazardous components. The work is interesting, and which can be published after a minor reversion by replying the following questions:
Response: We highly appreciate the reviewer’s valuable comments, positive approval and sincere contributions. We have carefully responded the following comments and accordingly revised the manuscript for significant improvement.
- In Introduction section: the authors should address previous Reviews involving MIT and TCM. Why is there still an empty space for you to write such a review?
Response: Thanks very much for the important guidance. According to the literature review, there is no comprehensive Reviews involving MIT applications for TCM researches especially in English. To some extent, our present review can fill an empty space.
- In Section 2: I suggest the authors to provide a schematic diagram to clear describe the principle of each method.
Response: Thanks very much for the valuable suggestion and kind instruction. The schematic diagram is added in Figure 1 to clear describe the principle of each method. (Updated Figure 1)
- In Sections 2 and 3: A table for summary of discussions might be better.
Response: Thanks. A table is now added for summary of discussions in sections 2 and 3. (added Table 2)
- Emulsion polymerization is mentioned in Section 2, but there is no corresponding section in the following text.
Response: Emulsion polymerization is now deleted from the text.
- The classification of different methods in Section 2must be reconsidered. I suggest: a. Polymerization Methods and New Imprinting Technologies and Strategies should be reviewed as two sections. And b. In New Imprinting Technologies and Strategies, I suggest the authors should be discuss in the following way: New imprinting methods (nano imprinting and surface imprinting); new template strategies (dummy temple and multiple temple); new monomer strategies (multifunctional monomer and stimulus response).
Response: Thanks very much for the valuable suggestion. As suggested, we have changed the classification of different methods in Section 2, in the revised manuscript, as follows, “2.2.1. New Imprinting Methods
2.2.1.1 Surface Imprinting Technology
2.2.1.2. Nanoimprinting Technology
2.2.2. New Template Strategies
2.2.2.1 Multitemplate Imprinting Strategy
2.2.2.2. Dummy Template Imprinting Strategy
2.2.3. New Monomer Strategies
2.2.3.1 Multifunctional Monomer Strategy
2.2.3.2. Stimulus Response Imprinting Strategy”
- When describing the disadvantages and advantages of each one, please address the disadvantages and advantages of each one.
Response: Thanks very much. We added the disadvantages and advantages of each new imprinting method. For example, Line 176-178, Line 189-191, Line 200-205, Line 208-215, Line 224-228, Line 236-241.
- Please add more your own critical ideas in the text.
Response: As suggested, we have added some critical ideas in the revised manuscript, as follows, “3) MIPs are rarely used in TCM preparations, and the application of stimulus response strategy should be strengthened in TCM slow and controlled release preparations, separation and purification, and formulation typing. The application of such pH-responsive nano-carrier might offer a potential platform for controlled delivery and increasing the bio-availability of drugs. (Line 779-783).
4) The MIPs’ function of selective recognition and targeted delivery/localization leads to perspective researches, namely, the early detection and treatment of related diseases using MIT in TCM. (Line 784-786).
6) MIPs are simple to prepare and easy to operate. If the laboratory preparation process can be successfully extended to industrial production, it will provide favorable conditions for its wider use. Since most of these technologies are still confined to the lab, pilot-scale investigations and tests are necessary to ensure their reproducibility and scalability in the real world. (Line 794-796).”.
some comments are added for related examples in red. As follows, “…MIP-SPE could be used as a useful tool for detecting and quantifying SD in a variety of TCM.” (Line 312-313)
“…The experimental results proved that DMIPs can be used to specifically separate and purify target products from some complex biological systems, and that the DMIPs may be used as a potential drug delivery system of compound herbal formulas.” (Line 494-497)
“In conclusion, the molecular imprinted polymer synthesized in the present study served to enrich and recover FA from biological samples and TCM extracts.” (Line 563-565)
“This work can inspire scientists to employ MOF and MIP as electrochemical sensing agents, which may play an important role in drug monitoring and drug quality control.” (Line 752-754)

Round 2
Reviewer 1 Report
The manuscript has improved a lot from the previous version. But I still suggest some comments for minor revisions. I think that the revised manuscript is substantially improved and recommended to accept after minor revision.
LINE 86 According to the literature review, there are few related reports on MIT for TCM study before 2017. Most reports have been increasingly published since 2017. The senctence can be added here.
LINE 204 Delete the hyperlink ‘for multiresidue analysis’.
LINE 256 Table 2 is better to be classified according to the chemical structure of analytes or the type of TCM real sample.
Author Response
Reviewer 1: Comments and Suggestions for Authors
The manuscript has improved a lot from the previous version. But I still suggest some comments for minor revisions. I think that the revised manuscript is substantially improved and recommended to accept after minor revision.
Response: We highly appreciate your positive approval, sincere contributions and continuous endeavors. We have carefully clarified the important points and accordingly revised the manuscript for improvement. In addition, we have added more methods for comparison according to your previous comments (update Table 1). Your valuable guidance also benefits greatly our future work.
- LINE 86 According to the literature review, there are few related reports on MIT for TCM study before 2017. Most reports have been increasingly published since 2017. The sentencecan be added here.
Response: Thanks very much for your kind professional revision. The sentence is now added in the revised manuscript. As follows, “According to the literature review, there are few related reports on MIT for TCM study before 2017. Most reports have been increasingly published since 2017. Therefore, in this work, the applications of MIT in the study of TCM are reviewed comprehensively, focusing on the recent advances since 2017.”. (Line 86-89)
- LINE 204 Delete the hyperlink ‘for multiresidue analysis’.
Response: Thanks very much for the considerate review. We have deleted the hyperlink ‘for multiresidue analysis’ in the revised manuscript. (line 205)
- LINE 256 Table 2 is better to be classified according to the chemical structure of analytes or the type of TCM real sample.
Response: Thanks very much for the valuable instruction/guidance. As instructed, we have added two columns to the leftmost of Table 2 according to the chemical structure of analytes and application aspects. (updated Table 2)